# Liquid Biopsy-Guided Interventional Oncology: A Proof of Concept with a Special Focus on Radiotherapy and Radiology

**DOI:** 10.3390/cancers14194676

**Published:** 2022-09-26

**Authors:** Natalia Malara, György Kovacs, Francesco Bussu, Teresa Ferrazzo, Virginia Garo, Cinzia Raso, Patrizia Cornacchione, Roberto Iezzi, Luca Tagliaferri

**Affiliations:** 1Department of Health Sciences, University Magna Grecia, 88100 Catanzaro, Italy; 2Nanotechnology Research Center, University Magna Grecia, 88100 Catanzaro, Italy; 3Gemelli-INTERACTS, Università Cattolica del Sacro Cuore, 88168 Rome, Italy; 4Department of Medical Surgical and Experimental Sciences, Sassari University, 07100 Sassari, Italy; 5UOC Radioterapia Oncologica, Dipartimento di Diagnostica per Immagini, Radioterapia Oncologica ed Ematologia, Fondazione Policlinico Universitario A. Gemelli IRCCS, 88168 Rome, Italy; 6UOC di Radiologia, Dipartimento di Diagnostica per Immagini, Radioterapia Oncologica ed Ematologia, Fondazione Policlinico Universitario A. Gemelli IRCCS, 88168 Rome, Italy

**Keywords:** liquid biopsy, radiotherapy, interventional oncology, CTCs, ct-DNA, minimal residual disease, cancer

## Abstract

**Simple Summary:**

Over the last decade radiology interventions for tumor ablation have steadily increased and new procedures of interventional Oncology (IO) have been developed. Minimal residual disease (MRD) management and its quantification has a critical role to assess the efficacy or failure of the treatment adopted.The assessment of the MRD depends by multiple and repetitive measurements performed through a no invasive procedure like as Liquid biopsy (LB). LB is a rapidly evolving technology with potentially relevant consequences in the management of patients with cancer. Despite its high potential, the LB approach cannot give spatial information and still lacks of standardization and required specificity and sensitivity to be extensively implemented in the routine clinical practice. The review is aimed at investigating how LB can monitor the effects of locoregional treatment highliting the potential synergies between interventional techniques and LB to improve cancer patient’ management. The review suggests combining Interventional techniques and LB technology could improve LB overcoming its sensitivity limitations. Furthermore, LB techniques may enhance the outcomes of minimally invasive loco-regional treatments not only via ongoing periodic MRD monitoring of a patient during and subsequent to IO procedures.

**Abstract:**

Although the role of liquid biopsy (LB) to measure minimal residual disease (MRD) in the treatment of epithelial cancer is well known, the biology of the change in the availability of circulating biomarkers arising throughout treatments such as radiotherapy and interventional radio-oncology is less explained. Deep knowledge of how therapeutic effects can influence the biology of the release mechanism at the base of the biomarkers available in the bloodstream is needed for selecting the appropriate treatment-induced tumor circulating biomarker. Combining existing progress in the LB and interventional oncology (IO) fields, a proof of concept is provided, discussing the advantages of the traditional risk assessment of relapsing lesions, limitations, and the timing of detection of the circulating biomarker. The current review aims to help both interventional radiologists and interventional radiation oncologists evaluate the possibility of drawing a tailor-made board of blood-based surveillance markers to reveal subclinical diseases and avoid overtreatment.

## 1. Introduction

Over the last decade, radiology interventions for tumor ablation have steadily increased, and new procedures have been developed. Radiofrequency, microwave, or cryotherapy are the most commonly used techniques supported by further progress in imaging tools such as ultrasound fusion and contrast, computed tomography (CT) fluoroscopy, magnetic resonance (MR) thermometry, and positron emission tomography PET guidance [1]. An additional alternative to thermal ablation is irreversible electroporation, which transiently disrupts cell membranes through electric pulses, killing cancer cells, although this technology is still in its early stages [2]. An expanded role in the future will deliver novel therapeutics into tumors through either catheter-based approaches or direct intratumor injections of chemotherapeutic agents or oncolytic viral therapies [2] and bacterial therapies [3]. Moreover, several ongoing treatment trials combine immunotherapy with interventional oncology (IO) to improve antigen presentation, which can be accomplished with ablative or catheter-directed therapies [3]. For infiltrative tumors or to locally control nonresectable lesions, which cannot be effectively accomplished by either external beam radiation therapy or standard thermal ablative techniques, a new procedure in interventional radiotherapy, called brachytherapy, combining an advanced imaging technique and an image navigator system, has been implemented. The integrated approach has the potential to shorten the acquisition time of diagnostic–therapeutic information and radioactive seed or catheter placement, promoting tumor control. Another emerging therapeutic procedure is high-intensity focused ultrasound ablation, which allows intratumor drug delivery through either selective opening of the vascular barrier or nanoparticle disruption [1].

In cancer patients showing an oligometastatic setting, the primary endpoint of IO is to achieve complete remission of the treated lesion and to reduce the incidence of distant spreading cells by controlling the risk of local recurrence [4]. This risk, together with the incidence of distant spreading, is strictly dependent on the number of residual cancer cells, locally and/or circulating, in the peripheral blood after treatment [5]. Although the goal of cytoreductive therapy is to eradicate all malignant cells, a substantial portion of patients develop resistant cancer cell clones, so-called minimal residual disease (MRD), detectable after therapy, which ultimately leads to cancer recurrence [6].

A critical milestone in MRD management is its quantification by multiple and repetitive measurements to detect the precise cut-off corresponding to the minimal threshold of detectable cancer cells of patients responding well to therapy and, on the other hand, that can be used as an end-point of the therapy when the values of the markers fall below the minimum detectable [7]. The MRD level, at any time, reflects the anticancer treatment response, as well as intratumor and systemic immunologic effects, having the potential to counteract cancer relapse. The deep analysis of MRD needs to include characterization of the high-risk features that antagonize the response to treatment. On the other hand, the type of treatment and the anatomical accessibility of the tumor would conditionate the MRD monitoring methods. Because of the differences in clinical practice, it has not yet been identified as a gold-standard method for quantifying MRD [8].

Liquid biopsy (LB) can indirectly diagnose the presence of MRD by revealing both quantitative and qualitative dynamic modifications of the circulating tumor cells (CTCs), circulating tumor DNA (ct-DNA), tumor-specific microRNA, and circulating exosomes [8]. These biomarkers are shed by the tumor in the bloodstream, with their entity and availability reflecting different tumor tissue stages, i.e., vascularization, vessel permeabilization, inflammation, and apoptosis or necrosis cell death [9]. Additionally, release of CTCs could be influenced by high proliferation, spontaneous detachment from the matrix, and molecular equipment that enables cancer cells to cross the endothelial wall, leading to cancer relapse [10].

All circulating biomarkers are increasingly being investigated to improve their pertinence, standardization, and reproducibility. LB can be considered a precious parameter in the therapeutic decision-making process, to monitor and detect MRD in patients with solid cancers eligible for interventional therapy, and also to select patients needing treatment adjustments to improve their quality of survival [8]. The promising clinical impact of molecular and cellular circulating markers consists of a personalized approach based on the control and monitoring of specific tumor biological key markers anticipating the development of the secondary lesion through quantitative and qualitative estimation before their clinical image-guided evidence (Figure 1).

## 2. Impact of Liquid Biopsy in Radiotherapy

As per the up-to-date state of the art, cancer cell clonal monitoring within the tumor tissue under the selective pressure of the treatment is dictated by anatomic imaging, conducting scans every few months, or testing blood tumor-associated antigens (carcinoembryonic antigen (CEA), carbohydrate antigen 19-9 (Ca19.9), etc.) at the systemic level [11]. To start the process, the first step is to obtain an informative biopsy before and over therapy time. A biopsy is considered “informative” when it shows high diagnostic yields, is less invasive, is well tolerated by patients, can include information about more biomarkers, and can provide an evaluation of the tumor microenvironment adaptations to therapeutic pressures. Although the anatomic approach has generally worked well, currently, interest is focused on gaining tissue informative biopsies able to guarantee molecular readouts of the most active cancer areas rather than those of quiescent or necrotic lesions. Targets are primarily chosen based on accessibility, surrounding structures, obstacle avoidance, and patient conditions (e.g., sedation requirements and pain control). Furthermore, numerous reports evaluating a large number of tissue biopsies have defined that harvesting multiple tissue cores does not increase complication rates [12], tissue sampling procedures remain invasive, and the standardization of biopsy materials procedures remains problematic. In an ideal setting, future biopsies will be performed by targeting “hot spots” identified by molecular imaging within the tumor lesion [13]. In recent years, the development of liquid biopsy methodology has introduced an alternative point of view for serially monitoring treatment response, MRD, and relapse. The quantification of MRD can be based on combining information from two main sources, liquid and tumor biopsy, to mine more accurate and informative data points. However, to optimize the liquid biopsy approach to unmask MRD-positive patients, it is reasonable to advocate that MRD testing would be performed by circulating biomarkers. Compliance between the treatment and the choice of MRD biomarker is a crucial step in designing a personalized monitoring platform, for example, to discriminate the molecular products released in peripheral blood or vesicles (exosomes, microvesicles, etc.), circulating cells of epithelial and/or mesenchymal coming from tumor tissue, and endothelial cells with different differentiation grades (Table 1).

Circulating DNA (ct-DNA) is the most widely analyzed tumor-related element during anticancer treatment, as it keeps the same genomic signatures in the matching tumor, and the analysis performed by next-generation sequencing technologies improves the rates of MRD quantification.

Ct-DNA provides data about the ability of some cancer cell subpopulations to fight elimination, revealing the competitive phenomena that undergo continuous remodeling at genetic levels during therapy. This approach allows the interrogation of clonally divergent, distant lesions without bias, allowing longitudinal monitoring of patients [14,15]. On the other side, ct-RNA in the bloodstream might be linked to epigenetic pathways of tumor plasticity, development, and progression. Both ct-DNA and ct-RNA are released in body fluid by tumor tissue as a consequence of cell death, necrosis, or apoptosis, or encapsulated in vesicles as secretion products by living cells [15]. These encapsulated vesicles are distinguished by size in microvesicles, apoptotic bodies, circulating exosomes, and tumor-educated platelets (TEPs) and are actively released in body fluids containing different molecules, nucleic acid fragments, proteins such as chemokines, and cytokines [16]. Their analysis can provide information on the dynamic interplay between cancer and immune cells throughout the treatments [17].

In conclusion, CTCs represent a heterogeneous population of epithelial, mesenchymal, epithelial–mesenchymal transition, endothelial, and reactive histocytes cells [18].

Real-time quantification and characterization of those populations can be considered the best goal in order to identify what happens inside the treated tumor for the following:CTCs correspond to the tumor progeny triggering, with a high probability, further lesions [19,20].Characterization and quantifications of CTCs correspond to the identification and quantification of specific MRD [21].In oligometastatic patients, the presence of CTCs in peripheral blood reflects the persistence of tumor lesions and their high probable access to the systemic bloodstream [22].In patients with multimetastatic disease, CTCs can be identified as a longitudinal scan of the heterogeneous cell clonality [23].The endothelial cell subtype, circulating endothelial cells (CECs), and endothelial progenitor cells (EPCs) provide useful information about tumor angiogenesis and when associated with cardiovascular parameters, as well as side effects due to anticancer treatment on heart function [18].

### 2.1. Minimal Residual Disease Assessment in Radiotherapy by Circulating Tumor DNA

Based on the biology of ct-DNA as free fragmented molecules in peripheral blood (cell-free-DNA or cf-DNA), its application in MRD surveillance during interventional radiology provides specific indications. For ct-DNA use in clinical practice, some limits must be considered. Very low levels must be detected within a total pool of cf-DNA (cell-free DNA), especially in pretreated cancer patients submitted to interventional oncology. Ct-DNA variations must be distinguished from those resulting from premalignant lesions such as age-related clonal hematopoiesis or due to the effect of chemotherapy on normal cells undergoing apoptosis. Moreover, it is necessary to consider the impact of biological activity (metabolic activity, cell death mechanisms, innate and adaptative immunity reaction) and anatomical factors (blood–tissue barrier permeabilization) on the dispersion of ct-DNA after local radiological treatment.

The detection of epigenetic alterations is also applicable to the analysis of ct-DNA carried out by polymerase chain reaction (PCR) or sequencing [14]. With increased accuracy and reduced sequencing costs, routine clinical implementation of ct-DNA analysis can be considered now more feasible and may achieve a greater impact on patients with radiotherapy-treated cancer.

Currently, the dose prescription or the decision between only radiotherapy or combined treatment with chemotherapy or surgery is determined by clinicopathological factors and is not influenced by biomarkers; therefore, some patients receive too much radiation or too many therapies, while others underestimate the risk of recurrence. The stratification of patients gained by LB allows to orient the therapy toward a more or less aggressive treatment. In the case of low levels of ct-DNA, the dose of radiotherapy can be reduced, or combined therapies can be avoided; instead, in patients with high levels of ct-DNA, it can lead to more aggressive therapeutic regimens (Figure 2). Ct-DNA profiling at baseline and Days 91, 181, and 361 was performed [24,25], emphasizing that long timing of longitudinal observation for this biomarker is needed after therapy.

Patients with high ct-DNA levels after neoadjuvant chemoradiotherapy and before surgery have very high relapse rates [24]. Furthermore, it must be considered that unchanged or increased ct-DNA levels during radiotherapy could indicate resistance to the treatment or progression of micrometastases, giving the chance to change the treatment strategy [25].

Emerging studies are showing promising performance of ct-DNA analysis for detecting MRD after radical radiotherapy [24,25,26,27,28]. In non-small-cell lung cancer (NSCLC). ct-DNA analysis of post-radiotherapy plasma samples achieved a 94% detection rate in patients who experienced recurrence [24].

Few studies use LB analysis to detect minimal/molecular residual disease in radiotherapy-treated patients; one factor impeding the conduct of prospective clinical trials in this field is the rapidly evolving technological platforms for the detection of ct-DNA and CTCs [24,25,26,27,28]. Many studies evaluating novel LB platforms in the MRD setting have been retrospective. Numerous other retrospective studies have suggested the role of LB-based surveillance after radical or neoadjuvant treatment. LB is also used to identify predictive biomarkers, which are mainly based on qualitative properties reflecting the biological performance of the tumor. Currently approved ones are related to molecular target therapy such as epidermal growth factor receptor (EGFR) tyrosine kinase in NSCLC [27].

Response prediction to radiotherapy, by molecular biomarkers, remains a useful research area to correct the use of radiotherapy, select suitable and unsuitable patients, and can adjust the dose prescription. Patients with oligometastatic disease could be candidates for more aggressive treatment incorporating local therapies [29,30]. A recent study in oligometastatic prostate cancer has uncovered the potential of LB as a predictive biomarker for response to advanced treatment [25] (Table 2).

There is also the possibility to combine ct-DNA with circulating proteins (prostate-specific antigen in prostate cancer) or with imaging (combination of ct-DNA and MRI imaging) [25,26,27] to improve the informative data setting linked to molecular biomarkers in MRD monitoring.

### 2.2. Minimal Residual Disease Assessment in Radiotherapy by Circulating Tumor Cells

Serial collection of CTCs during radiotherapy is useful to study the mechanism of treatment resistance in cancer types. CTCs can be grown in vitro [9,10,14] or subcutaneously engrafted into immunodeficient mice to provide a renewable source of patient tumor tissue for molecular profiling, genomic studies, and drug tests [18]. In patients with NSCLC treated with fractioned radiotherapy, it was hypothesized that the destruction of the tumor mass promotes the passage of tumor cells into the circulation, contributing to the formation of distant metastases [31]. The data do not support this hypothesis. The increased CTC positivity rate after radiotherapy could be due to an epiphenomenon depending on the actinic effects on the vessel’s permeabilization [32].

Real-time monitoring data indicated a statistically significant (compared to pretherapy levels) increase in the number of CTCs starting soon (10–15 min) after therapy. The increased number of CTC lasts on average 20 min before returning to the pretherapy level [32]. No significant changes were observed in CTC count in a time range of 8 and 16 h (hrs) after therapy crosswise all treatments. Current studies revealed an acute increase in CTCs in a time window consistent with fluctuations in tumor vascular permeability. Moreover, the short-lived increase in CTC concentration was followed by a notable absence of CTCs 8 h post-treatment, consistent with physiological suppression of blood flow in tumor tissue. This is indirectly confirmed by the return of CTCs count back to pretherapy levels 16 h after therapy when vessel occlusion effects are minimized. Combination therapy also had a similar trend; however, these changes were not as appreciable [33]. Aggregates of tumor and healthy cells and clusters of tumor cells are released into the bloodstream; these aggregates are intercepted by the lung or the spleen and rapidly eliminated from circulation, while individual cancer cells can remain in circulation for much longer periods (up to 30–90 min). Long-term effects were analyzed using tumor growth delay analysis and histopathology data in addition to the last CTCs scan on Day 21. By Day 21, following the initial treatment, radiation significantly delayed tumor growth and reduced the number of tumor cells in circulation. The therapeutic effect on tumor volume correlated well with the total CTC count at the end of the evaluation period [33,34,35]. After high-dose radiotherapy, lack of metastatic tumor mass following CTC release indicates low viability, clonogenicity, or both; on the other hand, real-time monitoring of CTCs provides a window on acute therapeutic events in real time, different from traditional approaches which can take months to detect changes associated with therapy [35].

Recently, a work demonstrated that at doses of 10 gray (Gy) or above, radiation disrupts the tumor vasculature leading to irreparable damage to nearby well-oxygenated tumor cells. While hypoxic tumor tissues are known to be radio- and chemoresistant; these data highlight the importance and relevance of adjuvant chemotherapy to control systemic disease with the use of vascular disrupting drugs in combination with radiation or not [36].

The decrease in CTCs after radiotherapy treatment was demonstrated by different studies and independently by the technology of analysis used. The Cell Search system demonstrated that 15% of patients are a positive CTC count before and 7% after treatment with Fluoro- EPISPOT assay 49% versus 36% and with Cell Collector 48% versus 39% [26,27,28,29] (Table 2).

CTC count is correlated with clinical course and response to treatment; patients who become CTC-negative after radiotherapy had a response rate of 86%, so CTC-positive after treatment is a poor outcome indicator. Similarly, a significant decrease in CTC number is found after radiotherapy in patients with localized NSCLC [27].

In prostate cancer (PC), prostate-specific antigen (PSA) serum level is not significantly different in patients with CTC-negative and positive samples, suggesting that the major source of PSA serum is not related to PSA—Secreting CTCs but rather to the tumor itself, independently from its transformation condition [34]. Several studies have reported significant correlations between the number of CTCs with patient survival and/or prostate cancer recurrence after treatment. These correlation studies have highlighted a clear trend towards a CTC value in correlation with the disease stage and towards a prognostic and predictive impact of CTC detection in prostate cancer [24,25,26,27,28,29,34].

Additionally, a significant correlation between CTC count and a patient’s body mass index (BMI) was found. The possible cause of this correlation is the high concentration of leptin, insulin, insulin-like growth factor-1, and low adiponectin, which together are likely to promote growth and progression. This observation underlies as in some types of tumors the metabolism induces a different behavior on CTCs availability [35].

### 2.3. Minimal Residual Disease Assessment in Radiotherapy by Circulating Exosomes

The secretion of cellular products is one of the direct effects that can be observed during radiotherapy. Irradiated tumor cells produce exosomes, which are released into the tumor microenvironment. The DNA damage triggers the mechanism of increased exosome secretion by radiation cells, mainly by the activation of the P53/TSAP6 axis and the Wnt pathway. Through exosome secretion and exchange, irradiated cells target non-irradiated cells promoting migration, radioresistance, and DNA damage by promoting cytokines secretion by dendritic cells in the tumor microenvironment. The content of exosomes produced and released into body fluids significantly changes composition after tumor radiation therapy. Exosomal loads mainly include exosomal miRNAs, proteins, and other substances, such as circRNA and lipids regulating the aggressive cancer phenotype. Exosomes after radiotherapy suppress aggressive phenotypes of tumor cells, such as proliferation and migration. On the other hand, after irradiation, exosome-mediated signaling [16].

## 3. The “Liquid Redundancy” of Interventional Oncology

As data reported in the literature suggest that LB can be used in monitoring radiotherapy outcomes, a few interventional oncology clinical trials use LB technologies with the same aim. Interventional radiology plays an integral role in delivering locoregional therapies for patients with both primary and metastatic disease. Both experimental treatment response monitoring and clinical follow-up may include tumor markers and CT, PET, and MRI. Microsatellite instability and mutational status, alpha-fetoprotein, and CA 19-9 are tumor markers that have been widely used for patients. Monitoring CTCs and ct-DNA may serve as a helpful additional tool for interventional radiologists to monitor patients before and after procedures to overcome the limited sensitivity and specificity of traditional markers.

Some studies have observed that mutational status assessed in tumor tissue predicts response to some minimally invasive therapies, including ablation and embolization [36,37,38], and can extend this concept to the mutational state of ct-DNA to improve the management of cancer patients. Furthermore, the biological response of tumor cells at the tissue level to irradiation is the result of the coexistence and interaction of five radiobiology factors (5R), and it is possible to hypothesize that these factors may affect one or more circulating biomarkers. The five radiobiological determinants of treatment outcome in external radiotherapy are the same determinants conditioning the brachytherapy outcome. Repair of sublethal-induced cellular/DNA damage, redistribution of the cells within the cell cycle, revascularization/reoxygenation of the surviving cells, matrix remodeling, growth, apoptosis, cell death, and the reaction of the immune response reassume the principal biological consequences of the local treatment [37,38,39]. Local effects during brachytherapy could be tracked in the blood. Conceptually, the typology of tumor products released in the blood could be influenced by the prevalent biological effect within the 5R, as graphed in Figure 3. The five biological effects described as 5R could be considered to describe the local effects induced by hypoxia during chemoembolization or transarterial chemoembolization (TACE), which combines embolization with the local delivery of chemotherapy [40]. The liquid redundancy of TACE effects could reiterate the same points of view graphed in Figure 3 for brachytherapy, with a difference in terms of intensity, duration, and fluctuation of the levels of the circulating markers considered. On these bases, the applications of liquid biopsy could extend to additional opportunities for interventional radiologists to use this technology in treatment selection and postprocedural treatment of patients with cancer.

## 4. Discussion

MRD has been identified as the single most robust prognosticator of overall and cancer-free survival in both cancer pediatric and adult patients. Serial testing of MRD is recommended, although robust trial evidences to validate the optimal timing and frequency are required.

In the oncology of solid cancer, LB offers the opportunity to measure MRD in molecular and cellular terms, opening a new alternative era to traditional follow-up. On the other hand, the possibility to choose the different types of circulating biomarkers, and the relative methodology of analysis, suggest the necessity to develop a proof of concept on the utility of LB to personalize MRD monitoring [6,11]. Interventional oncology, focused on local control of the lesion, has the option of choosing a good MRD marker within the LB panel. LB markers can be expressed in real time, especially if analyzed in combination with all tissue modifications, before, during, and after treatment. Starting from the permeabilization of the vessel in the first step of irradiation or thermal ablation until tumor remodeling over time, the increased permeabilization of tumor vessels and necrosis induced by local treatment generate a storm of molecules that diffuse into body fluids following tumor remodeling and cell death. After a variable number of days from the local intervention, studies have demonstrated agreement with the observation of the decrease in circulating biomarkers associated with local tumor size reduction [24,25,26]. One of the question points raised here is after how many days does the decrease or increase in the selected biomarker reflect the tumor size modification? Results reported by clinical trials focusing on LB, reported in Table 2, monitoring MRD after radiotherapy, emphasize that there are different timings of longitudinal observation when assessing the selected biomarker. All studies highlighted that after therapy, accidental dissemination of molecules and cells occurs in the proximal fluids of the tumor [33,34]. Their systemic diffusion includes molecular products from healthy cells of the surrounding tissues and the tumor. Thus, unintentional release determines a deep interference that increases the general analytical noise or disturbance, favoring mistakes in the estimation phase of the LB biomarkers. This condition suggests the need to respect the time of latency or postpone the analysis and estimation phase after it. However, clinicians should be informed of this particular timing and that the results of the analysis can be affected by a high risk of false-positive or false-negative events. This period of latency, between the first treatment and the biomarker collection, could be called “molecular sedimentation time” (MST), because it represents the time needed for the interfering factors to sediment or reduce their presence in the blood (Figure 4). The time required for the catabolism of the interfering factors, occurring through the function of the hepatic, renal, and immune emunctories, is variable and individual. However, it is possible to reduce the latency time by using a strong tumor-specific circulating biomarker. The high specificity of a marker can guarantee a lower rate of false interpretation. Therefore, the MST is inversely proportional to the specificity of the circulating biomarker. In this scenario, the waiting time between the most used LB biomarkers, as shown in the clinical trials (reported in Table 2), is lower for CTCs rather than cf-DNA. This difference is potentially due to the high specificity and sensibility of CTCs compared to the other circulating biomarkers in the LB panel.

CTCs can inform about the persistence of MRD and, from their characterization, also about the type or area of the tumor that restarts its proliferative behavior. Ct-DNA can provide information on clonal resistance in terms of mutations. Moreover, isolation of CTCs and short-time culture offer the opportunity to understand the survival of the prevalent clonal subset or the cause of the growing relapse through their direct sequencing (in situ hybridization, single-cell sequencing) or their immunocytochemical characterization. Recently [17,18], it has been demonstrated that the molecules secreted by cultured cells during in vitro short culture, the so-called secretome, are enriched by cytokines and growth factors. The characterization of the secretome, after IO, could represent an interesting point of view in the characterization of MRD to identify an integrative therapeutic approach. Moreover, the enhanced effect of IO for immunotherapy [19] found in the characterization of the secretome of CTCs is a further point to monitor in real time the interplay between tumor and immune system [17].

## 5. Conclusions

In the era of immunotherapy alternatives for MRD-directed therapy and treatment of frankly relapsed disease, it is yet imprecise which patients with MRD-positive disease will receive the most benefit from the chosen treatment or therapeutic consequences. It remains uncertain the importance of repeating MRD assessments in patients who contract MRD-negative disease after treatment. Therefore, before a patient relapses, a transition through an MRD-positive state is required, and this may be an opportunity for intervention. It is uncertain at this point whether achieving deeper molecular responses is clinically significant or if additional therapy produces deeper molecular responses. In the future, we need to focus on these questions to minimize exposure to the adverse effects associated with IO therapy, including cytokine release syndrome, as well as for important health and economic reasons. In the future, it is important that the complicated instrumentation used to detect and characterize circulating biomarkers [41,42] becomes less complicated by using lab-on-chip and point of care able to inform, in real time, clinicians about tumor dynamism [43,44] before and after an oncology intervention [45]. Moreover, the combination of LB and imaging can be addressed to improve the compliance of cancer patient management for reducing MRD and preventing tumor relapse. LB is a rapidly evolving technology with potentially relevant consequences in the management of patients with cancer, probably destined to modify oncological therapeutic strategies in the next decade.

Despite its high potential, the LB approach cannot provide spatial information and still lacks the standardization, specificity, and sensitivity required to be extensively implemented in routine clinical practice. Interventional oncologists should accept this new technology to potentiate the synergies between interventional techniques and LB to improve cancer diagnosis and therapy. Interventional techniques may help LB technology overcome sensitivity limitations. Furthermore, LB techniques may improve the application and enhance the outcomes of minimally invasive locoregional treatments.

## Figures and Tables

**Figure 1 cancers-14-04676-f001:**
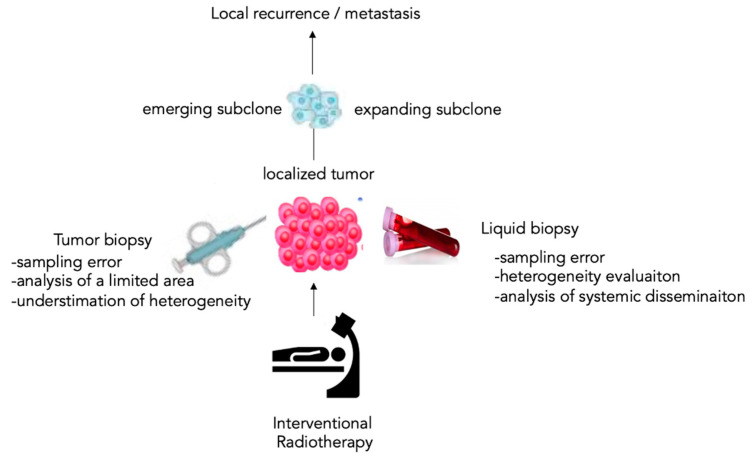
The liquid paradigm of the tumor biopsy. The liquid biopsy (LB) has the advantage of including the analysis of the circulating tumor products, representing the subclinical level of cancer disease systemic dissemination by unmasking emerging or expanding subclones involved in local recurrence or metastasis.

**Figure 2 cancers-14-04676-f002:**
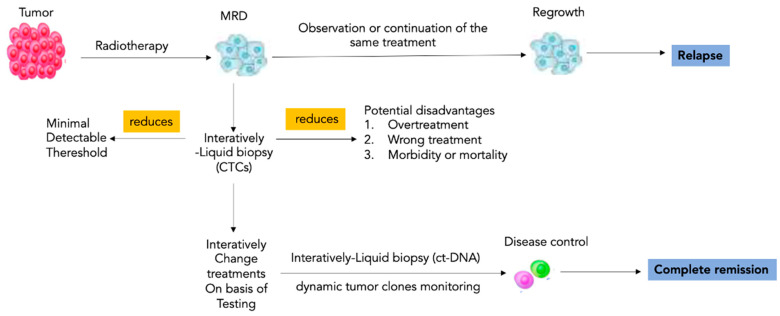
Impact of liquid biopsy in radiotherapy. Risk stratification through liquid biopsy allows care models to be personalized to the needs of patients within each cohort. LB-based longitudinal scans allow directing the radiotherapy toward a more or less aggressive treatment to modulate the radiotherapy dosage or to induce the choice of more aggressive therapeutic regimens. Interactive liquid biopsy has a key role to individuate and maintain the right tailored balance between the minimal detectable threshold of cancer cells and the emergent signs of overtreatment. Note: MRD, minimal residual disease; CTCs circulating tumor cells; ct-DNA, circulating tumor DNA.

**Figure 3 cancers-14-04676-f003:**
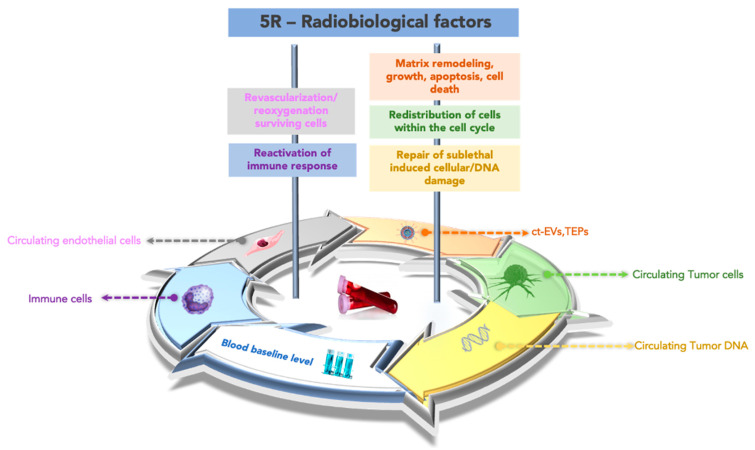
The “liquid redundancy” of the 5R biological factors. A proof of concept to individuate the circulating biomarkers linked to the prevalent biological factor, within the 5R, induced by interventional oncology treatment, influencing the relative effectiveness. Note: ct-RNA, circulating tumor RNA; ct-EVs, circulating vesicles TEPs, tumor-educated platelets.

**Figure 4 cancers-14-04676-f004:**
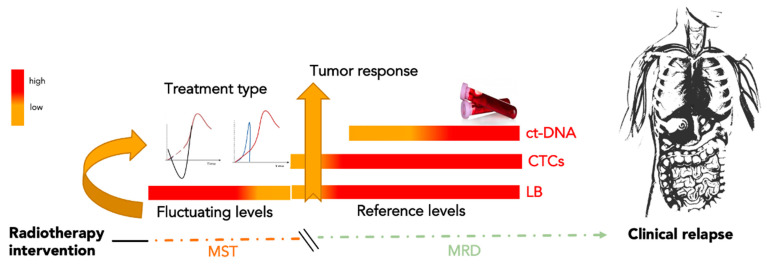
Molecular sedimentation time. Molecular sedimentation time (MST) is the period of latency between the first IO treatment and biomarker collection. During this period, interfering factors existing in the blood are produced prevalently by healthy tissue around the treated tumor lesion. Monitoring of minimal residual disease (MRD) begins with the first specific circulating tumor biomarker available. CTCs, being highly specific to tumor disease, are candidates to be a good marker for qualifying MRD by reducing the MST duration induced by IO. After MST, further combined analysis of CTCs and ct-DNA could represent a better surveillance approach for a longitudinal scan of MRD. Note: MST, molecular sedimentation time; MRD, minimal residual disease; ct-DNA, circulating tumor DNA; CTCs, circulating tumor cells, LB, liquid biopsy.

**Table 1 cancers-14-04676-t001:** Circulome.

Biomarkers	Indication	Advantage	Challenges	References
ct-DNA 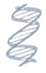	Treatment resistance evaluation	Established isolation procedure	-High degree of fragmentation-Low levels of ct-DNA in the amount of cf-DNA	Roy D. et al., 2021 [14]
ct-RNA 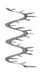	Prediction of response	Easy isolation	-Limited stability -RNAs may be tightly linked to EVs	Roy D. et al., 2021 [15]
EVs 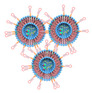	Prediction of response	Protect genomic from degradation	-Technically challenge in isolation procedure -Million of Evs are released every day by many cell types	Yixin Shi et al., 2022 [16]
CTCs 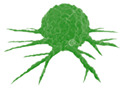	Minimal residual disease evaluation	Tumour specific information High degree of heterogeneity	-Technically challenge in standardization -Development of appropiates device for high throughput at low cost	Malara N. et al., 2018 [17]
CECs, EPCs 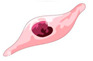	Treatment/overtreatment evaluation	Tumour angiogenesis information/side effects therapy	-Technically challenge in standardization	Lanuti P. et al., 2018 [18]

ct-DNA circulating tumor DNA, cf-DNA circulating celle free DNA, ct-RNA circulating tumor RNA, EVs extracellular veshicles, CTCs circulating tumor cells, CECs, Circulating endothelial cells, EPCs, Endothelial progenitor cells.

**Table 2 cancers-14-04676-t002:** Landscape clinical trial employing liquid biopsy in radiotherapy.

Clinical Trial Name and Number	Goal	Sample Size	Liquid Biopsy	Source Reference Number
ORIOLE trial NCT02680587	Evaluated the effect of SABR on hormone-sensitive oligometastatic prostate cancer	54 randomized patients	Stereotactic ablative radiation ct-DNA	Phillips R. et al. 2020 [29]
RAVENS trial NCT04037358	Evaluated progression-free survival of men who have HSOPCa after randomization to SABR versus SABR and radium-223	64 randomized patients	Enumeration of CTCs at baseline and day 181. ct-DNA profiling at baseline, Days 91, 181, and 361.	Hamza Hasan et al. 2020 [30]
SABR-COMET-3 trial NCT03862911	Compare the effect of SABR versus standard of care (SOC); in patients with 1–3 oligometastatic with a controlled primary tumor of any solid tumor histology	297 randomized Patients	Enumeration of CTCs and ct-DNA in order to evaluate the correlation between oligometastatic disease and oncological outcomes (at baseline, at 3 months, and at disease progression or at the end of the study)	Robert Olson et al. 2020 [31]
SABR-COMET-10 trial NCT03721341	Compare the effect of SABR versus standard of care; in patients with 4–10 oligometastatic with a controlled primary tumor of any solid tumor histology	159 randomized patients	Enumeration of CTCs and ct-DNA in order to evaluate the correlation between oligometastatic disease and oncological outcomes	David A Palma et al. 2019 [32]
OPVIDO-SBRT-phase 2	Compare nivolumab versus nivolumab + radiation therapy in patients with advanced inoperable melanoma.	20 patients with inoperable or metastatic melanoma.	Detect mutations of BRAF and NRAS on ct-DNA	Nora Sundhl et al. 2019 [33]
ICE-PAC trial ACTRN12618000954224	Evaluate the efficacy and safety of the avelumab with stereotactic ablative body radiotherapy (SABR) in mCRPC	31 patients	Analyses of cf-DNA and cf-RNA (AR splice variants AR-V7 and AR-V9, recognized for their strong association with pathogenicity)	Edmond M. Kwan et al. 2021 [34]

Note: SABR, stereotactic ablative radiotherapy; ct-DNA, circulating tumor DNA; HSOPCa, hormone-sensitive, oligometastatic prostate cancer; CTCs, circulating tumor cells; cf-RNA, cell-free RNA; AR, androgen receptor; mCRPC, metastatic castration-resistant prostate cancer.

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
