# Peer review of "Liquid Biopsy-Guided Interventional Oncology: A Proof of Concept with a Special Focus on Radiotherapy and Radiology"

_cancers, 2022, doi:10.3390/cancers14194676_

Round 1

Reviewer 1 Report

This review discussed the advantages and disadvantages of the traditional risk assessment based circulating biomarker in liquid biopsy and interventional oncology fields. This review has certain guiding significance for clinical management. More importantly, the points addressed here are useful for interventional radiologists and interventional radiation oncologists. After a few minor points were further addressed, this review could be accepted in Cancers.

1. The image resolution of Figures is too poor. And some sentences have too many Spaces

2. Liquid biopsy (LB) contains circulating tumor cells (CTCs), circulating tumor DNA (ctDNA), tumor-specific microRNA and exosomes. The roles of these circulating biomarkers, especially exosomes, in cancer radiotherapy are preliminary reviewed in Section 2, and thus need more deep-discussion. For example, the applications of exosome in radiotherapy mainly include predicting radiotherapy efficacy, predicting tumor prognosis, and optimizing the regimen of tumor treatment (doi: 10.31083/j.fbl2707205).

3. The combinations of radiotherapy/interventional radiooncology with other therapeutic strategies should be discussed, such as cancer immunotherapy. Some newly-published reports could be cited, such as doi: 10.1186/s12885-022-09820-w, doi: 10.3390/pharmaceutics14071466, doi: 10.1007/s11060-019-03363-0 and doi: 10.1016/j.canlet.2015.01.009

Author Response

Paper: Manuscript ID: cancers-1888446

Title: "Liquid Biopsy Guided Interventional Oncology: a Proof of Concept with a Special Focus on Radiotherapy and Radiology”

Dear Editor,

We are pleased to submit to Cancer, a revised version of the manuscript entitled “Liquid Biopsy Guided Interventional Oncology: a Proof of Concept with a Special Focus on Radiotherapy and Radiology”, and a detailed response to the comments of the Reviewers. The MS has been revised to comply with the observations of the Reviewers. Specifically:

o          we have corrected typos in the MS and we improved English and style of the MS. Where necessary, we have rewritten words or sentences to convey more clearly the message of the paper.

o          we have partially re-written the MS

o          we have changed the resolution images

Considering the corrections made to the manuscript, and the positive comments of the Reviewers on the work, we hope that the paper can be accepted for publication in this present form. We warmly thank the Reviewers that, with their comments, contributed to improve the work.

In the following, you will find the original comments from the reviewers (bold black text) and the point-by-point response of the authors (bold blue text). In the manuscript all modifications are tracked (yellow text)

With Regards

The authors 

Reviewers: 1

Comments to the Author

Reviewer 1

R1. The image resolution of Figures is too poor. And some sentences have too many Spaces

Author

A1: The authors will begin by thanking the reviewer for his constructive comments, the image resolution was increased to > 300 DPI and the spaces removed

R2. Liquid biopsy (LB) contains circulating tumor cells (CTCs), circulating tumor DNA (ctDNA), tumor-specific microRNA and exosomes. The roles of these circulating biomarkers, especially exosomes, in cancer radiotherapy are preliminary reviewed in Section 2, and thus need more deep-discussion. For example, the applications of exosome in radiotherapy mainly include predicting radiotherapy efficacy, predicting tumor prognosis, and optimizing the regimen of tumor treatment (doi: 10.31083/j.fbl2707205).

A2: The authors thank the reviewer for this clarification and the reference indicated was useful to improve our knowledge about this issue. To this regard, a new paragraph titled: “Minimal residual disease assessment in radiotherapy by circulating exosomes”, was added

R3. The combinations of radiotherapy/interventional radiooncology with other therapeutic strategies should be discussed, such as cancer immunotherapy. Some newly-published reports could be cited, such as doi: 10.1186/s12885-022-09820-w, doi: 10.3390/pharmaceutics14071466, doi: 10.1007/s11060-019-03363-0 and doi: 10.1016/j.canlet.2015.01.009

A3: The authors thank the reviewer for the indication and the references were very useful to improve our knowledge about this issue and one of these articles was introduced in the MS. At the same time, the authors believe that the implications of radiotherapy treatment in the context of immunotherapy deserve a dedicated discussion. The purpose of this review is to sensitize the reader to the usefulness of using liquid biopsy-based monitoring of minimal residual disease after radiotherapy. The referring circulating markers of the effects of an immunotherapy approach are different and do not necessarily include the circulating markers discussed here.

Reviewer 2 Report

We do agree that surgeon should be considering before metastasis is found the which MRD-positive finding is required, and this may be a critical factor for surgical consideration.
However, LB can be the most promising personal precision care in oncology care when it accepted by most care providers and cost-effective as common biomarkers. 

Author Response

Paper: Manuscript ID: cancers-1888446

Title: "Liquid Biopsy Guided Interventional Oncology: a Proof of Concept with a Special Focus on Radiotherapy and Radiology”

Dear Editor,

We are pleased to submit to Cancer, a revised version of the manuscript entitled “Liquid Biopsy Guided Interventional Oncology: a Proof of Concept with a Special Focus on Radiotherapy and Radiology”, and a detailed response to the comments of the Reviewers. The MS has been revised to comply with the observations of the Reviewers. Specifically:

o          we have corrected typos in the MS and we improved English and style of the MS. Where necessary, we have rewritten words or sentences to convey more clearly the message of the paper.

o          we have partially re-written the MS

o          we have changed the resolution images

Considering the corrections made to the manuscript, and the positive comments of the Reviewers on the work, we hope that the paper can be accepted for publication in this present form. We warmly thank the Reviewers that, with their comments, contributed to improve the work.

In the following, you will find the original comments from the reviewers (bold black text) and the point-by-point response of the authors (bold blue text). In the manuscript all modifications are tracked (yellow text)

With Regards

The authors 

Reviewers: 2

Comments to the Author

We do agree that surgeon should be considering before metastasis is found the which MRD-positive finding is required, and this may be a critical factor for surgical consideration.

However, LB can be the most promising personal precision care in oncology care when it accepted by most care providers and cost-effective as common biomarkers.

A1- The authors strongly thank the reviewer for this comment

Reviewer 3 Report

Recommendation: Publish after major revisions noted.  

Comments:  

This manuscript highlights liquid biopsy guided interventional oncology. The authors need to address the following comments and revise the manuscript accordingly. 

  1. This manuscript is in need of substantial editing, English language and style improvement.
  2. Abstract: Page 1, line 27: Consider to rewrite the aim of this article.
  3. Introduction: Page 2, line 78: Please consider emphasizing each biomarker for MRD monitoring and including references for each biomarker. Add a few sentences about the use of LB in the cancer continuum, as well as the following references. a) Roy D, Lucci A, Ignatiadis M, Jeffrey SS. Cell-free circulating tumor DNA profiling in cancer management. Trends Mol Med. 2021 Jul 23:S1471-4914(21)00182-9. doi: 10.1016/j.molmed.2021.07.001. b) Lin, N.; Lin, Y.; Xu, J.; Liu, D. et al. A multi-analyte cell-free DNA–based blood test for early detection of hepatocellular carcinoma. Hepatol Commun. 2022;00:1–11. c) Roy, D.; Tiirikainen, M. Diagnostic Power of DNA Methylation Classifiers for Early Detection of Cancer. Trends Cancer 20206, 78–81. DOI:https://doi.org/10.1016/j.trecan.2019.12.006. d). Lin S, Gregory RI. MicroRNA biogenesis pathways in cancer. Nat Rev Cancer. 2015;15(6):321–33.
  4. Consider to highlight exosomal microRNAs and add the following references. a) Roy, D.; Pascher, A.; Juratli, M.A.; Sporn, J.C. The Potential of Aptamer-Mediated Liquid Biopsy for Early Detection of Cancer. Int. J. Mol. Sci. 2021, 22, 5601. https://doi.org/10.3390/ijms22115601.  b) Wu L, Zhou WB, Zhou J, Wei Y, Wang HM, Liu XD, Chen XC, Wang W, Ye L, Yao LC, Chen QH, Tang ZG. Circulating exosomal microRNAs as novel potential detection biomarkers in pancreatic cancer. Oncol Lett. 2020 Aug;20(2):1432-1440. doi: 10.3892/ol.2020.11691. Epub 2020 May 30. PMID: 32724386; PMCID: PMC7377032.
  5. Table1: Consider to add references corresponding to each biomarker.
  6. Page 4, line 150: Consider paraphrasing this section.
  7. Page 7, line 274: Table 2. Consider adding two columns that emphasize the techniques/assays utilized in each trial as well as the study findings.
  8. Figure 3: Increase the front size within the figure. Consider not using background colors for the boxes.
  9. Figure 4: Increase the scale bar's front size.

Author Response

Paper: Manuscript ID: cancers-1888446

Title: "Liquid Biopsy Guided Interventional Oncology: a Proof of Concept with a Special Focus on Radiotherapy and Radiology”

Dear Editor,

We are pleased to submit to Cancer, a revised version of the manuscript entitled “Liquid Biopsy Guided Interventional Oncology: a Proof of Concept with a Special Focus on Radiotherapy and Radiology”, and a detailed response to the comments of the Reviewers. The MS has been revised to comply with the observations of the Reviewers. Specifically:

o          we have corrected typos in the MS and we improved English and style of the MS. Where necessary, we have rewritten words or sentences to convey more clearly the message of the paper.

o          we have partially re-written the MS

o          we have changed the resolution images

Considering the corrections made to the manuscript, and the positive comments of the Reviewers on the work, we hope that the paper can be accepted for publication in this present form. We warmly thank the Reviewers that, with their comments, contributed to improve the work.

In the following, you will find the original comments from the reviewers (bold black text) and the point-by-point response of the authors (bold blue text). In the manuscript all modifications are tracked (yellow text)

With Regards

The authors 

Reviewer 3

Recommendation: Publish after major revisions noted. 

Comments: 

This manuscript highlights liquid biopsy guided interventional oncology. The authors need to address the following comments and revise the manuscript accordingly.

R1.       This manuscript is in need of substantial editing, English language and style improvement.

A1- The authors thank the reviewer for his constructive comments. The MS was revised and a substantial editing was performed.

R2.      Abstract: Page 1, line 27: Consider to rewrite the aim of this article.

A2- The authors thank the reviewer for this indication and the sentence was rewritten

R3.      Introduction: Page 2, line 78: Please consider emphasizing each biomarker for MRD monitoring and including references for each biomarker. Add a few sentences about the use of LB in the cancer continuum, as well as the following references. a) Roy D, Lucci A, Ignatiadis M, Jeffrey SS. Cell-free circulating tumor DNA profiling in cancer management. Trends Mol Med. 2021 Jul 23:S1471-4914(21)00182-9. doi: 10.1016/j.molmed.2021.07.001. b) Lin, N.; Lin, Y.; Xu, J.; Liu, D. et al. A multi-analyte cell-free DNA–based blood test for early detection of hepatocellular carcinoma. Hepatol Commun. 2022;00:1–11. c) Roy, D.; Tiirikainen, M. Diagnostic Power of DNA Methylation Classifiers for Early Detection of Cancer. Trends Cancer 2020, 6, 78–81. DOI:https://doi.org/10.1016/j.trecan.2019.12.006. d). Lin S, Gregory RI. MicroRNA biogenesis pathways in cancer. Nat Rev Cancer. 2015;15(6):321–33.

A3- Each biomarker for MRD monitoring was added, as requested, in the corresponding Table 1

R4.      Consider to highlight exosomal microRNAs and add the following references. a) Roy, D.; Pascher, A.; Juratli, M.A.; Sporn, J.C. The Potential of Aptamer-Mediated Liquid Biopsy for Early Detection of Cancer. Int. J. Mol. Sci. 2021, 22, 5601. https://doi.org/10.3390/ijms22115601.  b) Wu L, Zhou WB, Zhou J, Wei Y, Wang HM, Liu XD, Chen XC, Wang W, Ye L, Yao LC, Chen QH, Tang ZG. Circulating exosomal microRNAs as novel potential detection biomarkers in pancreatic cancer. Oncol Lett. 2020 Aug;20(2):1432-1440. doi: 10.3892/ol.2020.11691. Epub 2020 May 30. PMID: 32724386; PMCID: PMC7377032.

A4- The authors thank the reviewer for this indication to improve our knowledge about this issue and one of the references indicated was introduced in the MS.

R5.      Table1: Consider to add references corresponding to each biomarker.

A5- As reported above, the Table 1 was modified as requested

R6.      Page 4, line 150: Consider paraphrasing this section.

A6- Thank you for this indication and the section was re-written

R7.      Page 7, line 274: Table 2. Consider adding two columns that emphasize the techniques/assays utilized in each trial as well as the study findings

A7- The authors thank the reviewer for his indication. However, the table 2 has not been modified, because the methodology used for the detection of circulating markers are the conventional ones and not defining an effective difference. On the other hand, in paragraph 2 of the review, the methodology was briefly reported for each biomarker

R8.      Figure 3: Increase the front size within the figure. Consider not using background colors for the boxes

A8- The authors thank for these indications. The figures were modified as requested

R9.      Figure 4: Increase the scale bar's front size

A9- as reported above the figures were modified as requested

Round 2

Reviewer 3 Report

This version of the manuscript is improved. Please publish.